# Does the Change in the Diagnostic Criteria for Gestational Diabetes in Poland Affect Maternal and Fetal Complications? A Prospective Study

**DOI:** 10.3390/medicina58030398

**Published:** 2022-03-07

**Authors:** Edyta Cichocka, Janusz Gumprecht

**Affiliations:** Department of Internal Medicine, Diabetology and Nephrology in Zabrze, Medical University of Silesia, 40-055 Katowice, Poland; jgumprecht@sum.edu.pl

**Keywords:** change in criteria, complications, gestational diabetes mellitus, insulin therapy

## Abstract

*Background and objectives:* Gestational diabetes mellitus (GDM) is a significant risk factor of maternal and fetal complications. The aim of the study was to compare two groups of patients with GDM treated in 2015/2016 (Group-15/16), and in 2017/2018 (Group-17/18) and to answer the question whether the change in the diagnostic criteria for GDM affected maternal and fetal complications. *Materials and Methods*: A retrospective analysis was conducted. The study included 123 patients with GDM (58 patients/Group-15/16 and 65 patients/Group-17/18). *Results:* No significant differences were found between the groups. In Group-17/18, GDM was significantly more often diagnosed based on fasting glycemia (33.8%) compared with Group-15/16 (22.4%; *p* = 0.000001). GDM was significantly more often diagnosed based on 2-h oral glucose tolerance test (OGTT; 44.8%) compared with Group-17/18 (29.2%; *p* = 0.000005). In Group-15/16, insulin was started in 51.7% of patients compared with 33.8% in Group-17/18 (*p* = 0.04287). Despite more frequent insulin therapy in Group-15/16, insulin was started later (30th week of gestation) and significantly more frequently in older patients and those with higher BMI values compared with Group-17/18 (27th week of pregnancy). The number of caesarean sections and spontaneous deliveries was also similar in both periods. No difference was found in the prevalence of neonatal complications, including neonatal hypo-glycemia, prolonged jaundice or heart defect. In addition, no differences were found between the parameters in newborns. *Conclusions:* The change in the criteria for the diagnosis and treatment of GDM translated into the mode of diagnosis and currently it is more often diagnosed based on abnormal fasting glycemia. Currently, a lower percentage of patients require insulin therapy. However, less frequent inclusion of insulin may result in higher postprandial glycemia in the third trimester of pregnancy in mothers, thus increasing the risk of neonatal hypoglycemia immediately after delivery.

## 1. Introduction

Gestational diabetes mellitus (GDM), which is defined as a disorder of carbohydrate metabolism that occurs during pregnancy, is one of the most common pregnancy-related metabolic complications [1]. Predisposition to GDM is present before pregnancy, and hormonal changes and weight gain, which are typical of pregnancy, lead to the manifestation of GDM [1,2]. Environmental factors, including particularly the obesity epidemic and higher age of pregnant women, are also involved in the manifestation of the disease [3]. The prevalence of hyperglycemia during pregnancy increases rapidly with maternal age [2]. Previous treatment of infertility is also another important risk factor for hyperglycemia during pregnancy. In this group of patients, almost half of pregnancies (48.9%) were complicated by hyperglycemia, despite the relatively low maternal age (<30 years) [1,4]. GDM occurs in 1%–25% of patients depending on the country of origin, which is related to different diagnostic criteria and the influence of ethnic factors [1,5]. It is caused by pancreatic β-cell dysfunction which is also present before pregnancy and is manifested by physiologically increasing gestational insulin resistance. This dysfunction is conditioned by different factors (i.e., a specific genetic disorder, mild autoimmune destruction, or other processes resulting in β-cell damage).

Diagnosis of GDM and implementation of appropriate management allow the reduction in the risk of complications in neonates and perinatal mortality rates and improve maternal and offspring prognosis. GDM is a strong predictor of metabolic disorders in the offspring, both in childhood and adulthood. The criteria for the diagnosis of GDM and the glycemic targets for patients have changed in recent years. In 2005, Diabetes Poland (Polish Diabetes Association) adopted 100 mg/dL in the venous blood plasma as the upper limit of normal fasting glucose in the general population and in pregnant women. According to the Diabetes Poland and the Polish Society of Gynecologists and Obstetricians, fasting plasma glucose of 100 mg/dL and the value of 140mg/dL after a 2-h 75-g glucose tolerance test were the diagnostic criteria for GDM [6]. Until 2014, GDM was diagnosed in Poland based on the abovementioned test.

Since 2013 [7], in accordance with the guidelines of the World Health Organization (WHO), hyperglycemia during pregnancy has been divided in Poland into two categories:-diabetes mellitus during pregnancy—a disorder which meets the general criteria for diabetes mellitus already at diagnosis-gestational diabetes mellitus—it is mostly manifested during the second or third trimester of pregnancy and meets the specific criteria for the pregnancy period.

Diagnosis of diabetes mellitus during pregnancy is established when the following general conditions for the diagnosis of diabetes are met:-fasting blood glucose ≥ 126 mg/dL (7 mmol/L) or-2-h plasma glucose ≥ 200 mg/dL (11.1 mmol/L) following a 75 g oral glucose load or-random plasma glucose ≥ 200 mg/dL (11.1 mmol/L) in the presence of diabetes symptoms.

In turn, the diagnosis of GDM is established if one or more of the following criteria are met:-fasting blood glucose 92–125 mg/dL (5.1–6.9 mmol/L) or-1-h plasma glucose ≥ 180 mg/dL (10.0 mmol/L) following a 75 g oral glucose load or-2-h plasma glucose 153–199 mg/dL (8.5–11.0 mmol/L) following a 75 g oral glucose load.

Since 2014, GDM has been diagnosed in Poland based on the abovementioned test.

Since 2017, there have been changes in the glycemic targets for self-monitoring, which are recommended for patients treated for carbohydrate metabolism disorders diagnosed during pregnancy [8].

Before 2017, the following glycemic targets for self-monitoring were recommended: fasting glucose between 60–90 mg/dL and the 1-h postprandial glucose level <120 mg/dL. Since 2017, the following values have been recommended: fasting blood glucose between 70–90 mg/dL, and 1-h after the start of the meal <140 mg/dL.

The value of maximum postprandial glycemia at 1-h was increased to 140 mg/dL (previously 120 mg/dL) to reduce the risk of hypoglycemia. The change in the fasting glycemic target was related to the publication of the results of the hyperglycemia and adverse pregnancy outcome study (HAPO study), which showed a continuous association between fasting glucose concentration and the fetal growth/fetal adverse outcomes if fasting blood glucose was >90 mg/dL in women who were not diagnosed with GDM based on the previous criteria. There was also a positive but weaker correlation between increasing glucose concentrations and maternal complications.

Relaxation of glycemic targets, especially postprandial glycemia, allowed many pregnant women to avoid insulin therapy and enabled them to continue dietary treatment until delivery.

## 2. Objective of Study

The aim of the study was to compare two groups of patients with GDM who were treated in the Diabetes Outpatient Clinic of the Clinical Hospital No. 1 in Zabrze, Poland and to answer the question of whether the relaxation of glycemic targets in patient treatment translated into obstetric outcomes of the compared groups, the prevalence of maternal and fetal complications, and the need to start insulin therapy. Two groups of patients were compared, i.e., patients treated in 2015 and 2016 (Group-15/16) and those treated in 2017 and 2018 (Group-17/18).

## 3. Patients and Methods

It was a retrospective study. The medical records of GDM patients of the Diabetes Outpatient Clinic of the Clinical Hospital No. 1 in Zabrze treated between 2015 and 2018 were analyzed. All patients diagnosed with GDM on the basis of the current criteria were trained in glucose self-monitoring (control of fasting glucose and glucose 1-h after the start of the meal) and monitoring of ketone bodies in urine in the morning during the first visit in the Diabetes Outpatient Clinic. Dietary training related to the principles of proper nutrition was also conducted (with the recommendations of a balanced diet containing an average of 25–30 kcal/kg, depending on the initial body weight before pregnancy). Visits took place every 4 weeks on average. However, they were more frequent when (fasting or postprandial) target glycemic values were higher for 7 consecutive days. Insulin therapy was initiated if glycemic targets for self-monitoring were higher for 7 consecutive days.

Part of the data was collected from the medical records, and patients were contacted by phone in the absence of information about delivery. The analysis of medical records including 123 patients was conducted (58 patients from Group-15/16 and 65 patients from Group-17/18). Patient characteristics of the study groups are presented in Table 1.

Statistical analysis was performed using Statistica 13 for Windows (StatSoft) and Rstudio software. For variables with non-normal distribution, data were presented as a median with lower (Q 1%–25%) and upper (Q 3%–75%) quartiles. The distribution of variables was assessed with the Shapiro Wilk test. The Mann-Whitney U-test was used to compare the variables. The relationships between the qualitative variables were assessed using the Pearson Chi-square test or Fischer test for 2 × 2 tables with a small number of cases. *p* Values < 0.05 were considered statistically significant.

## 4. Results

The study did not show significant differences in terms of the assessed parameters between the groups of pregnant women (Group-15/16 vs. Group-17/18). In both groups, patients had similar body weight, height, and body mass index (BMI) before pregnancy and maternal age at conception. During pregnancy, they achieved similar weight gain (9.5–10 kg) (Table 1).

The week when the diagnosis of GDM was established and the week of delivery were also similar in both groups. In over 50% of patients, GDM was related to the first or second pregnancy in both groups (Table 2). No differences were found between the parameters in newborns in terms of birth weight, body length or the Apgar score at 1 min (Table 1). The number of caesarean sections (30 patients—52% in Group-15/16 vs. 36 patients—56% in Group-17/18) and spontaneous deliveries (27 patients—48% in Group-15/16 vs. 30 patients—44% in Group-17/18) was similar in both periods (NS).

No differences were found between the groups in terms of the order of pregnancy in which GDM occurred (Table 2).

The comparison of neonates with a body weight of >4000 g showed no difference between the groups (body weight > 4000 g was found in 8.6% of neonates who were born in 2015 and 2016 compared with 6.15% in 2017 and 2018) (NS—Table 3).

No differences were found between cesarean section and macrosomia. There were five newborns with body weight > 4000 g in Group-15/16 (three deliveries by caesarean section) and four newborns in Group-17/18 (three deliveries by caesarean section). We also analyzed the basis on which the diagnosis of GDM was established and showed that GDM was significantly more often diagnosed based on fasting glucose in Group 17/18 (*n* = 22; 33.8%) compared with Group-15/16 (*n* = 13; 22.4%; *p* < 0.001). GDM was significantly more often diagnosed based on a 2-h 75 g oral glucose tolerance test (OGTT) in Group-15/16 compared with Group-17/18 (*n* = 26; 44.8% vs *n* = 19; 29.2%; *p* < 0.0001). Furthermore, in Group-15/16, GDM was significantly more often diagnosed based on two factors (fasting glucose and 2-h 75g OGTT) compared with Group-17/18 (*p* < 0.001). In both groups, single cases of GDM diagnosed based on two other abnormalities were found (Table 4).

In terms of GDM treatment, insulin was started significantly more often in Group-15/16 compared with Group-17/18 (*n* =30; 51.7% vs *n* = 22; 33.8%; *p* = 0.04244). In turn, in Group-17/18, dietary treatment was sufficient in 66.2% (*n* = 43; *p* = 0.04244; (Table 5, Figure 1).

Furthermore, in Group-15/16, insulin was significantly more often used in older patients (*p* = 0.019224) and in those with higher BMI values (*p* = 0.037626), which was not found in Group-17/18 (Table 6).

A difference was observed with a trend towards significance (*p* = 0.06) related to the week of pregnancy in which insulin was started, i.e., at the 30th (27; 33) week of pregnancy in Group-15/16 and the 27th (19; 31) week of pregnancy in Group-17/18.

Maternal comorbidities were compared between the groups. Hypothyroidism, which required L-thyroxine treatment, was significantly more prevalent in Group-17/18 compared with Group-15/16 (*n* = 18 vs. *n* = 7; *p* = 0.02596). No difference was found in the prevalence of arterial hypertension or polycystic ovary syndrome (PCOS) (Table 7).

Neonatal complications were also assessed. No difference was found in the prevalence of neonatal hypoglycemia, which required medical attention, prolonged jaundice, and heart defect (Table 8).

Neonatal hypoglycemia which required medical intervention was defined as that which required the administration of intravenous glucose in the neonate. Prolonged neonatal jaundice was defined as hyperbilirubinemia (>10 mg/dL) which lasted more than 14 days. In our group of neonates, other complications such as shoulder dystocia or the respiratory distress syndrome were not reported (most information was collected by phone interview).

## 5. Discussion

Since 2010, based on the results of the hyperglycemia and adverse pregnancy outcome (HAPO) study, the International Association of the Diabetes and Pregnancy Study Groups (IADPSG) has recommended the change in the diagnosis of GDM based on the three-point 75 g OGTT and new glycemic thresholds [9]. The diagnostic criteria for GDM proposed in the HAPO study were based on their predictive values for unfavorable pregnancy and obstetric outcomes. The proposal presented by IADPSG was accepted by many scientific societies worldwide, including Diabetes Poland. Currently, 75 g OGTT is a standard which is used to diagnose carbohydrate metabolic disorders in pregnancy.

Many studies have revealed different relationships between particular OGTT criteria and GDM-related complications. It is assumed that fasting glycemia is the strongest predictor for the subsequent development of diabetes in women after delivery [10] and the development of fetal macrosomia [11]. These data may prove particularly important in educating women with past GDM, making them aware of the extent of the problem, and encouraging them for regular glycemic control. In turn, 1-h glucose values have shown a significant relationship with obstetric complications [12,13]. When glucose level was not assessed 1-h after a 75 g oral glucose load, GDM was not detected in 18% of women with large-for-gestational-age fetuses [12].

The relaxation of self-monitored target glucose levels 1-h after the start of the meal and the change from 120 to 140 mg/dL resulted in avoiding too-frequent insulin therapy and obtaining similar obstetric outcomes with the use of dietetic treatment only. The study also showed that more than 50% of patients required insulin therapy at the time when “stricter criteria” were applied and only 33% of patients from Group-17/18. The acceptable self-monitored glucose levels in 2015 and 2016 translated into the need for more frequent initiation of prandial insulin therapy and basal insulin (target fasting glucose level <90 mg/dL) in the subsequent years, which had an influence on the percentage of insulin therapy in the study groups.

As presented above, despite more frequent insulin therapy in Group-15/16, it was initiated later (30th week of pregnancy on average), and significantly more often in older patients and in those with higher BMI values compared with Group-17/18 (27th week of pregnancy on average). This can be explained by the fact that patients from Group-15/16 who required insulin therapy mostly presented with predominant pancreatic β-cell dysfunction, which generally increases with age and depends on body weight [5,14,15]. One of the causes of less frequent start of insulin therapy in Group-17/18 was related to the fact that patients from this group were slimmer before pregnancy compared with those from Group-15/16. However, no statistically significant differences were found.

In Group-15/16, prandial insulin was started significantly more often, which means that patients had higher recommended postprandial glycemic levels (120 mg/dL; self-monitoring) (Table 5 and Figure 1). In Group-17/18, insulin was started significantly less often and basal insulin was mostly initiated (Table 5). The authors believe that less frequent inclusion of prandial insulin in Group-17/18 was related to the change in the postprandial glycemic threshold to 140 mg/dL.

The mode of treatment of GDM based on the modified criteria for glycemic control may have an influence on neonatal complications. Admittedly, no difference was found in the prevalence of neonatal hypoglycemia in our study.

We know that prevalent of neonatal hypoglycemia may be the result of higher postprandial glucose levels during pregnancy. Cutting the umbilical cord results in the disruption of maternal glucose transport, which in turn leads to rapid lowering of blood glucose in neonates. It can be particularly intense (transient hyperinsulinemia) in newborns of women with GDM, especially in those with higher postprandial blood glucose levels compared with target glucose concentrations [16,17].

In Group-17/18, insulin therapy was significantly less often initiated (33%), which may have indicated higher glycemic levels in the third trimester of pregnancy and an increased risk of neonatal hypoglycemia [18,19]. Again, the authors stress that higher postprandial glycemic levels in Group-17/18 were due to the change in the postprandial blood glucose threshold to 140 mg/dL compared with Group-15/16.

To conclude, the change in the criteria for the diagnosis and treatment of GDM translated into the mode of diagnosis and currently it is more often diagnosed based on abnormal fasting glucose levels. It also translated into the lower percentage of patients requiring insulin therapy. In turn, less frequent inclusion of insulin may translate into higher postprandial glucose concentrations in the third trimester of pregnancy in mothers, thus increasing the risk of hypoglycemia in the newborn immediately after delivery.

The limitation of the study is related to the small size of patient groups that were compared, and the results particularly concerning neonatal complications should be interpreted with great caution. Of note, since patients with GDM are mostly treated in diabetes outpatient clinics only until delivery, obtaining complete data related to labor or neonatal complications is not always possible and in our study such data were mostly collected by phone interview.

Regardless of the current criteria for treatment of GDM, the management of pregnant patients always requires a multidirectional approach, which includes regular self-monitoring of glucose level and of weight gain as well as monitoring of intrauterine fetal growth.

## 6. Conclusions

The relaxation of the criteria for GDM control did not significantly affect the obstetric outcomes of the groups of patients. The course of pregnancy in both groups was similar. The percentage of pregnancies with delivery of neonates >4000 g did not differ significantly. No differences were found in the anthropometric parameters in newborns and neonatal complications.

On the other hand, the change in the criteria for the diagnosis of GDM significantly affected the modes of diagnosis and treatment. In Group-15/16, GDM was significantly more often diagnosed based on 2-h OGTT (75 g glucose load; cut-off point at 140 mg/dL) and in Group-17/18 based on fasting glucose (cut-off point at 92 mg/dL).

In connection with lower values of self-monitoring of postprandial blood glucose in the in women with GDM, insulin therapy was started significantly more often, which was also shown in our study. More than 50% of patients required the initiation of insulin therapy in Group-15/16 compared with only 33% in Group-17/18. In addition, in Group-15/16, it was shown that insulin therapy depended on maternal age and BMI values and was more often introduced in older patients and in those with higher BMI values. However, this relationship was not confirmed in Group-17/18. Insulin therapy was started later (30th week of gestation) in Group-15/16 compared with Group-17/18 (27th week of gestation). A trend towards significance was observed (*p* = 0.06).

## Figures and Tables

**Figure 1 medicina-58-00398-f001:**
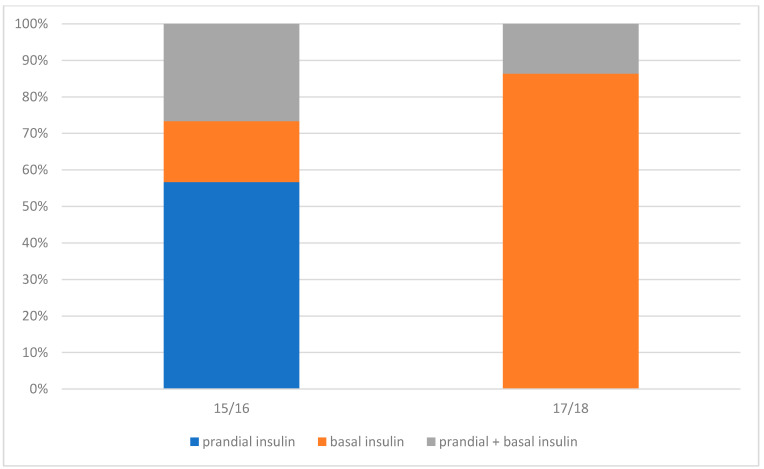
Mode of insulin therapy during pregnancy in groups 15/16 and 17/18.

**Table 1 medicina-58-00398-t001:** Characteristics of the study groups.

Variable	GDM in Group-15/16 (*n* = 58)			GDM in Group-17/18 (*n* = 65)			*p*
	Median	Q1	Q3	Median	Q1	Q3	
Body weight before pregnancy, kg	77.00	59.50	82.85	64.10	57.00	88.00	0.6974
Height, cm	165.00	160.00	170.00	165.00	160.00	170.00	0.9574
BMI, kg/m^2^	26.821	21.57	31.06	24.53	21.25	30.11	0.5900
Age, years	30.00	27.00	34.00	31.00	27.00	34.00	0.3750
Body weight gain during pregnancy, kg	10.00	6.00	15.00	9.50	6.00	14.50	0.9505
Week in which GDM was diagnosed	26.00	23.00	27.00	26.00	20.00	28.00	0.5397
Week in which insulin was started	30.00	27.00	33.00	27.00	19.00	31.00	0.0663
Week of delivery	39.00	37.00	40.00	39.00	36.00	40.00	0.5880
Neonatal body weight at delivery, g	3330.00	2810.00	3650.00	3250.00	2920.00	3500.00	0.7619
Neonatal body length at delivery, cm	53.00	52.000	56.00	54.00	52.00	56.00	0.9783
Apgar score at 1 min	10.00	9.00	10.00	10.00	9.000	10.00	0.7869

BMI—Body mass index, GDM—Gestational diabetes mellitus; Q1—lower (25%) quartile; Q3—upper (75%) quartile.

**Table 2 medicina-58-00398-t002:** Order of pregnancy in which Gestational diabetes mellitus (GDM) occurred.

Oder of Pregnancy	GDM in Group-15/16		GDM in Group-17/18		*p*
1	34	58.62%	36	55.38%	0.4197
2	15	25.86%	16	24.62%	
3 and the following pregnancies	9	15.52%	13	20.00%	
Total	58		65		

GDM—Gestational diabetes mellitus.

**Table 3 medicina-58-00398-t003:** Comparison of neonatal body weight at delivery.

GDM Group	<4000 g	>4000 g	*n*	GDM Group	<4000 g	>4000 g	*p*
15/16	53	5	58	15/16	91.40%	8.60%	*p* = 0.5455
17/18	61	4	65	17/18	93.85%	6.15%	
Total	114	9	123				

GDM—Gestational diabetes mellitus.

**Table 4 medicina-58-00398-t004:** Mode of diagnosis of gestational diabetes mellitus (GDM).

OGTT	Group—15/16		Group—17/18		*n*	*p*
Fasting	13	22.4%	22	33.8%	35	*p* < 0.0001
Fasting glucose and 1-h after OGTT	1	1.7%	4	6.15%	5	
Fasting glucose and 2-h after OGTT	18	31%	3	4.61%	21	
2-h after OGTT	26	44.8%	19	29.2%	45	
1-h and 2-h after OGTT	0	0	7	10.76%	7	
1-h after OGTT	0		7	10.76%	7	
Fasting and 1-h and 2-h after OGTT	0		3	4.6%	3	
Total	58		65		123	

OGTT—Oral glucose tolerance test.

**Table 5 medicina-58-00398-t005:** Mode of treatment of gestational diabetes mellitus (GDM).

Mode of GDM Treatment	GDM Group					
	15/16 (*n* = 58)		17/18 (*n* = 65)		*n*	
**Diet + Insulin**	30	51.7%	22	33.8%	52	*p* = 0.04244
Prandial insulin only	17	56.7%	0	0		
Basal insulin only	5	16.7%	19	86.4%		
Prandial+basal insulin	8	26.7%	3	13.6%		
**Diet**	28	48.3%	43	66.2%	71	

GDM—Gestational diabetes mellitus.

**Table 6 medicina-58-00398-t006:** The relationship between the mode of treatment and the age of patients and body mass index (BMI) values.

**Variable**	**GDM in Group-15/16** **Treated with Diet and Insulin**			**GDM in Group-15/16** **Treated with Diet Only**			** *p* **
	**Median**	**Q1**	**Q3**	**Median**	**Q1**	**Q3**	
BMI, kg/m^2^	28.05	24.30	31.97	24.53	20.51	25.60	*p* = 0.037626
Age, years	32.0	29.0	36.0	29.0	26.0	32.0	*p =* 0.019224
	**GDM in Group-17/18** **Treated with Diet and Insulin**			**GDM in Group-17/18** **Treated with Diet Only**			** *p* **
	**Median**	**Q1**	**Q3**	**Median**	**Q1**	**Q3**	
BMI, kg/m^2^	29.67	21.3	35.12	23.4	20.9	27.34	*p* = 0.125727
Age, years	31.5	29.0	34.0	30.0	27.0	35.0	*p =* 0.579396

BMI—Body mass index, GDM—Gestational diabetes mellitus.

**Table 7 medicina-58-00398-t007:** Maternal and neonatal comorbidities.

Maternal Comorbidities	GDM Group-15/16 (*n* = 58)	GDM Group-17/18 (*n* = 65)	Total	
Arterial hypertension	2	6	8	*p* = 0.19422
Hypothyroidism	7	18	25	*p* = 0.02596
Polycystic ovary syndrome	2	2	4	*p* = 0.64637

GDM—Gestational diabetes mellitus.

**Table 8 medicina-58-00398-t008:** Neonatal comorbidities.

Neonatal Comorbidities	GDM Group-15/16	GDM Group-17/18	Total	
Prolonged jaundice	9	10	19	*p* = 0.58926
Hypoglycemia	0	3	3	*p* = 0.14434
Heart defect	0	2	2	*p* = 0.27722

GDM—Gestational diabetes mellitus.

## Data Availability

Not applicable.

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
