# Peer review of "Does the Change in the Diagnostic Criteria for Gestational Diabetes in Poland Affect Maternal and Fetal Complications? A Prospective Study"

_medicina, 2022, doi:10.3390/medicina58030398_

Round 1
Reviewer 1 Report
Gestational diabetes is one topic of high importance. Despite the fact that is a retrospective study in a single center, the manuscript is interesting. However, in order to improve the manuscript I suggest the following :
-the title does reflect the content but in a precise way as is stated in Instructions for authors “The title of your manuscript should be concise, specific and relevant.”
-the abstract: it is not clear , you may “ Place the question addressed in a broad context and highlight the purpose of the study”
The text is good . Gestational diabetes is associate with large-for-gestational-age status and macrosomia, preeclampsia , shoulder dystocia , and neonatal morbidities, such as hypoglycaemia, hyperbilirubinemia, and respiratory distress syndrome. You discuss about cesarean section and vaginal delivery , but instrumental deliveries are associated with macrosomia. I suggest doing a comparison for all of these complications.
Author Response
Dear Reviewer,
Thank you for your review and comments. According your suggestion, I changed the title and added the question addressed in a broad context and highlight the purpose of the study. I compared also instrumental deliveries with macrosomia.
I highlighted all changes in yellow. I hope that now is ok.
Reviewer 2 Report
In this study Cichocka and Gumprecht compared the GDM diagnosis in two groups of patients with GDM treated in 2015/2016 and in 2017/2018. They found that GDM was significantly more often diagnosed based on fasting glycemia and 2-h OGTT. Moreover, they found no difference in the prevalence of neonatal complications concluding that less frequent inclusion of insulin may result in higher postprandial glycemia in the third trimester of pregnancy.
The manuscript is generally well written and only few changes are needed. In particular:
- The formatting style of timing must be uniformed in the whole manuscript. For example, lines 233-237, 1 hour is written in 3 different types (1-hour, 1 hour, 1h).
- An accurate revision of typing errors is recommended.
Author Response
Dear reviewer,
Thank you for your revision. According your suggestion I corrected the formatting style of timing in the whole manuscript and performed revision of typing.
Round 2
Reviewer 1 Report
In my opinion, the authors don’t answer properly to my suggestions. The title does suggest a proposal to modify the interpretation of the level of glycemia for gestational diabetes screening- a retrospective or a comparative study. The abstract remains unclear and in Material and Method, we cannot find cases with neonatal complications you analysed. Of course, your sample is too small for this analysis and I understand there were no cases with shoulder dystocia, respiratory distress syndrome, etc because the information was collected by phone interview.
Author Response
Dear Reviewer,
Thank you very much for your suggestions and valuable comments, which I followed and which have definitely improved the quality of the manuscript. As I mentioned in the Results section, the study limitation is connected with a relatively small group of patients and the lack of access to all the data related to neonatal complications. In the future, we obviously plan to extend the study on a larger cohort of patients.
Kind regards,
Edyta Cichocka